# Synergistic Antibacterial Effects of Gallate Containing Compounds with Silver Nanoparticles in Gallate Crossed Linked PVA Hydrogel Films

**DOI:** 10.3390/antibiotics13040312

**Published:** 2024-03-29

**Authors:** John Jackson, Claudia Helena Dietrich

**Affiliations:** 1Faculty of Pharmaceutical Sciences, University of British Columbia, 2405 Wesbrook Mall, Vancouver, BC V6T1Z3, Canada; 2Department of Pathology and Laboratory Medicine, University of British Columbia, Vancouver, BC V6T1Z7, Canada; claudinha.dietrich@gmail.com

**Keywords:** gallate, antibacterial, polyvinyl alcohol, synergy, silver nanoparticles

## Abstract

Currently available silver-based antiseptic wound dressings have limited patient effectiveness. There exists a need for wound dressings that behave as comfortable degradable hydrogels with a strong antibiotic potential. The objectives of this project were to investigate the combined use of gallates (either epi gallo catechin gallate (EGCG), Tannic acid, or Quercetin) as both PVA crosslinking agents and as potential synergistic antibiotics in combination with silver nanoparticles. Crosslinking was assessed gravimetrically, silver and gallate release was measured using inductively coupled plasma and HPLC methods, respectively. Synergy was measured using 96-well plate FICI methods and in-gel antibacterial effects were measured using planktonic CFU assays. All gallates crosslinked PVA with optimal extended swelling obtained using EGCG or Quercetin at 14% loadings (100 mg in 500 mg PVA with glycerol). All three gallates were synergistic in combination with silver nanoparticles against both gram-positive and -negative bacteria. In PVA hydrogel films, silver nanoparticles with EGCG or Quercetin more effectively inhibited bacterial growth in CFU counts over 24 h as compared to films containing single agents. These biocompatible natural-product antibiotics, EGCG or Quercetin, may play a dual role of providing stable PVA hydrogel films and a powerful synergistic antibiotic effect in combination with silver nanoparticles.

## 1. Introduction

The optimal features of a wound dressing are to provide an immediate anti-infective environment in a long-lasting hydrogel that may be easily removed from the wound by rinsing when required. Poly vinyl alcohol (PVA) has been extensively studied as a potential wound dressing material since it is biocompatible and may be provided as a drug-loaded, thin flexible film that quickly swells in water to form a hydrogel. The material is inexpensive and is used in existing commercial medical orthopedic devices [1,2]. PVA is generally available with high (99%) or low (under 90%) degrees of hydrolyzation which renders the material almost insoluble and fully soluble, respectively. For wound dressing applications, some control of the degradation rate of the PVA is preferred, leading many workers to describe crosslinking methods to prevent PVA dissolution including the use of borates [3], citric acid [4] or glutaraldehyde [5,6]. Our group previously investigated heat crosslinking of PVA (88% hydrolyzed) in the presence of silver nitrate or the simple ratio blending of 99% and sub 90% hydrolyzation level PVA polymers [7,8] to control degradation. Such PVA films allow for reasonable control of swelling and degradation while allowing the encapsulation and release of anti-infective silver salts or nanoparticles.

Silver is a well-established antibacterial agent used in numerous wound dressings. Silver sulfadiazine creams (prescription or online orders) have been used topically for many years and silver containing dressing materials such as Acticoat ^tm^ (silver nanoparticles) or Aquacel Ag ^tm^ are applied for longer-term treatment of wounds. Despite widespread use globally, the efficacy of these wound dressings is limited [9,10]. The improved antibacterial effect of nanoparticulate silver over ionic silver is now well established with a mechanism of action that may be additive when used with other antibiotics, especially in drug-resistant settings [11]. This additive or synergistic antibiotic effect against gram-positive and -negative bacteria of silver nitrate or silver nanoparticles in combination with existing antibiotics is well established [12,13,14,15]. Despite these known effects, there still remains a need for dual-drug-loaded anti-infective wound dressings, especially as drug resistant problems increase in clinics.

Gallate-containing molecules such as Tannic acid, Quercetin, and epigallocatechin gallate (EGCG) are polyphenols found in many plants and are known to have mild antibiotic effects against both gram-positive and -negative bacteria with minimal inhibitory concentrations determined in this laboratory of the order of 100 μg/mL as compared to silver nanoparticles at 2.5 μg/mL. The gallate-containing molecule Tannic acid, used for centuries to stabilize collagen in hides to produce leather, has been shown to create extensive hydrogen bonds with 99% hydrolyzed PVA to strengthen the polymer [16,17,18]. Similarly, unpurified tea polyphenol extracts which contain gallates have also been used with PVA in both a strengthening and antibacterial role [19,20,21]. Coincidentally, gallates (including Tannic acid, tea polyphenols, Quercetin, and EGCG) may act as “green” reducing agents to convert silver salts into stronger anti-infective silver nanoparticles [22,23,24,25,26,27,28,29,30,31,32,33]. This allows for the green and inexpensive creation of silver nanoparticles within gallate-containing, PVA cast films for a possible dual antibacterial drug composition for the treatment of wounds.

Although there is generally an improved antibiotic effect from using the dual-loaded compositions (silver with Tannic acid [23,24], Quercetin [29,33], and a minor effect with EGCG [34]), these previous studies usually show minor levels of inhibition and are complicated by the absence of compositions with silver nanoparticles alone. Almost all “green” synthesis studies do not allow for studies with silver nanoparticles alone since they are made in situ with the gallates present. Furthermore, most workers report the gallates form a stabilizing coating on the surface of the silver nanoparticles (e.g., Tannic acid [22,25], Quercetin [27,28,29,30], and EGCG [31] which may inhibit the normal antibiotic effect of the silver nanoparticles [26,31].

To overcome this problem of “green” synthesis gallate-capping of silver nanoparticles, we have investigated the use of premanufactured silver nanoparticles (AGNP) incorporated into gallate-containing PVA (88% hydrolyzed) cast films. By using fully water-soluble (88% hydrolyzed) PVA, the possible controlled crosslinking and aqueous degradation of the PVA films was explored. The antibacterial effects (against gram-positive and -negative bacteria) of all agents (gallate alone, silver nanoparticles alone, or gallate with silver nanoparticles) was measured using 96-well plate FICI synergy assays. Furthermore, for PVA films containing agents, CFU counting methods following 24 h incubations of bacteria with swollen films were used.

## 2. Materials and Methods

Poly (vinyl alcohol) (Selvol 540, 88 mole% hydrolyzed, Mw~150,000) was purchased from Sekisui Specialty Chemical Company (Dallas, TX, USA). Silver nanoparticles (10 nm Biopure-citrate) were purchased from Nanocomposix (San Diego, CA, USA). Tannic acid, Quercetin, glycerol, and silver salts (>99.0%) were purchased from Sigma-Aldrich (St. Louis, MO, USA). Epigallocatechin-gallate (EGCG > 94%) was purchased as a product, Teavigo, from DSM (Cambridge, ON, Canada). All solvents and other chemicals were purchased from Fisher scientific. Deionized water was used in the preparation of all formulations.

### 2.1. Manufacture of PVA Films

#### Film Preparation (Solvent-Cast PVA)

PVA solutions were prepared as a 2.5% *w/w* stock solution by slowly adding PVA powder to rapidly stirred water, followed by continued stirring and heating to 95 °C for approximately 60 min. When a clear solution had formed, the contents were cooled. Solutions with silver nanoparticles were prepared by adding known volumes of the stock solution to 20 mL volumes of PVA solution with or without glycerol (20% final concentration to PVA). Films were cast in 60 mm × 15 mm disposable polystyrene Petri dishes. The solutions in Petri dishes were left in a 37 °C oven overnight, in order for the water to evaporate. All dried films were stored in a dark cupboard before evaluation. Films were easily removed from Petri dishes with forceps after the rim coating on the vertical side of the Petri dish was cracked.

PVA films contained 500 mg of PVA and 100 mg of glycerol (where applicable). EGCG and Tannic acid films contained 0.04% Silver (to PVA) or 200 μg of silver, and Quercetin films contained 0.02% silver (to PVA) or 100 μg of silver. Silver salt solutions were prepared at 10 mg/mL concentrations in water and stored covered with aluminum foil in a dark cupboard until required.

### 2.2. Swelling Determinations

#### Film Swelling Studies

Film sections weighing 50 mg were placed on moistened 0.45 um filter discs (S Pak HA membrane 47 mm diameter, Millipore, Billerica, MA, USA) and weighed. The films and filters were covered with approximately 0.5 mL of water. After set time periods, the filter discs and adherent PVA-gallate gel were moved to a Millipore vacuum apparatus and vacuum was applied to draw all excess water from the filter over approximately 5 s. This was enough to remove all surface and loose water but not shrink the gels. The combined PVA gel and filter were reweighed and then placed back in water. The weight gain (termed swelling) was then calculated as a percentage of the original dry film weight.

### 2.3. Drug Release Studies and Characterization

Films (100 mg) containing EGCG, Tannic acid, or Quercetin along with silver nanoparticles were placed in 10 mM Hepes buffer (pH 7.3) (5 mL) and all this buffer was removed at regular intervals for silver analysis by Inductively Coupled Plasma (ICP Agilent, Santa Clara, CA, USA) analysis and gallate analysis by HPLC analysis. The 5 mL of Hepes buffer was then replaced with 5 mL of fresh buffer added to the films. Experiments were performed using triplicate samples.

Silver calibration standards (10 to 2000 ng/mL) were run every 30 samples. The ICP instrument displayed reproducible standard curves, over 75 sequential rounds of silver analysis with detection limits as low as 10 ng/mL. Each release study was run in triplicate for at least two weeks and the results plotted as the calculated percentage of silver released as a function of time. A Waters Acquity HPLC system (Milford, MA, USA) with Empower software version 1 and UV/VIS analysis was used for both Quercetin and EGCG quantitation. For EGCG, the mobile phase was a gradient of acidified water (glacial acetic acid 0.5%) and acetonitrile from 90:10 to 80:20 over 6 min followed by 6 more min with the 80:20 mobile phase and detection at 293 nm. For Quercetin, the mobile phase was isocratic using 0.5% acetic acid and acetonitrile at a 65:35 ratio and detection at 375 nm.

### 2.4. Bacterial Studies

The bacteria used were Methicillin-resistant *Staphylococcus aureus* (MRSA (USA 300)) (gram-positive) and *Escherichia coli* (*E. coli.* K12) (gram-negative) and were grown in lysogeny broth. These bacteria represent difficult-to-treat bacteria from the gram-positive and -negative categories are the go-to standard bacteria used by most groups in the microbiological field.

### 2.5. 96-Well Plate Checkerboard Assays

EGCG, Quercetin, or silver (expressed as the concentration of silver, not salt) were serially diluted 2-fold (using silver along one axis and EGCG or Quercetin along the other) across the 96-well plate. This was then followed by the addition of 100 uL of bacterial culture with an OD_600_ of 0.0025 to all wells. Plates were then wrapped with foil and incubated for 24 h at 37 °C. The turbidity in each well was then analyzed using a microplate reader at OD_600_. Control lanes containing drugs alone and no bacteria were also run to check for background interference.

### 2.6. Fractional Inhibitory Concentration Index (FICI) Determination

The FICI was calculated using values of the turbidity of the wells adjusted to background interference. The FICI of each agent (e.g., Silver or EGCG) was determined as the minimal inhibitory concentration (MIC) of one agent divided by the MIC of the other agent to the MIC of that agent alone. FICI was then computed as the sum of each agent’s FIC. The FICI values were then interpreted as follows: FICI ≤ 0.5, synergy; FICI 0.5–≤0.75 partial synergy; FICI 0.75–≤1.0, additive effect; FICI >1.0–≤4.0, indifference; and FICI > 4.0, antagonism as similarly described by others [35,36,37].

### 2.7. Colony Forming Unit, Kill-Curve Test

PVA films (12.5 mg) were placed in 20 mL flat-bottomed screw cap glass vials. Then, 1 mL of bacterial culture at OD_600_ of 0.005 in Lysogeny broth (LB) medium was added to the film which then swelled. The vials were left in the dark at 37 °C for 24 h. Samples of 100 uL were extracted from each tube at 24 h followed by 10-fold serial dilutions in small tubes. Then, 10 uL of each dilution was taken and pipetted onto LB agar plates. These plates were then incubated for 24 h at 37 °C. Colonies were counted using a low magnification optical microscope and the counts presented in log CFU/mL.

### 2.8. Statistics

The students T test (unpaired) was used to determine significance with a *p* value of less than 0.05.

## 3. Results

PVA films containing Tannic acid or EGCG at lower concentrations were generally clear and strong. With the addition of glycerol, the films were very flexible but did not break under pressure. Films containing Quercetin were only clear at low concentrations (up to 100 mg in 500 mg PVA) but dried films were cloudy after that. The cloudiness in films probably arose from the reducing solubility of Quercetin, EGCG, and Tannic acid in the drying water cast film so that as the water evaporated, the agents precipitated. The final composition of the major films (termed EGCG 75, EGCG 100, Quercetin 100, and Quercetin 200) is shown in Table 1.

### 3.1. Swelling Studies

PVA films with or without glycerol and just silver or no drugs dissolved almost immediately and fully in less than two hours. Films were manufactured using 500 mg of PVA (20 mL of 2.5% *w/v*) with various amounts of Tannic acid, EGCG, or Quercetin between 50 mg and 500 mg. All films were made with or without the plasticizing agent glycerol. Swelling data (at 5 min, 1 day and 4 days) are shown for all films containing glycerol in Table 2. Because these studies included 20 samples, only single samples were used and measured over multiple time points. The experiments were repeated twice and demonstrated the same concentration-dependent crosslinking effect of the gallates on PVA.

More detailed examples of the time courses of swelling for Tannic acid, EGCG, and Quercetin (all 100 mg in 500 mg PVA) are shown in Figure 1 and Figure 2.

Films containing Tannic acid swelled rapidly (15 min) to levels between 200 and 300% which remained swollen for 3 to 4 h before slowly breaking down over 24 h. The was little impact from the concentration of Tannic acid in the PVA or the inclusion of glycerol in these swelling studies (Figure 1).

Swelling for EGCG-loaded PVA films was similarly largely unaffected by the inclusion of glycerol and was more robust and long-lasting than that observed for Tannic acid. When 100 mg of EGCG in 500 mg PVA was used, swelling was stable at approximately 300% for 14 days (Figure 1). Using 50 mg of EGCG brought high initial levels of swelling that did not last, and when 200 to 500 mg of EGCG was used, swelling levels were constant over 4 days at around 100% (Table 2).

For Quercetin, swelling levels were initially higher but soon dropped down to stabilize in the 200 to 400% swelling range at 4 days (Table 2 and Figure 2). Quercetin could not be loaded uniformly in PVA films at levels higher than 300 mg in 500 mg PVA.

### 3.2. Drug Release Studies

Release studies were performed with PVA films containing concentrations of EGCG at 75 mg or 100 mg or Quercetin at 100 mg or 200 mg (to 500 mg of PVA) containing glycerol (20%) with or without silver nanoparticles. These films showed prolonged swelling times in the swelling studies (Table 2). Tannic acid loaded films were not studied because swelling did not prolong past 1 day. These films (EGCG 75 and 100 or Quercetin 100 and 200) were considered the most clinically relevant and release studies were only performed at 37 °C, also to match clinically relevant conditions.

EGCG released from films with a burst phase over one day followed by a more sustained release over the next six days (Figure 3). All samples reached approximately 40% of encapsulated EGCG released by day 7. The 75 mg EGCG no-silver sample released a little over 50% of drug but the release data were not significantly different to the other samples. Interestingly, one residual film for the 75 mg EGCG no-silver sample was broken down (polytron homogenizer with ethanol extraction) and showed 47% of the initially loaded EGCG still present in agreement with the data in Figure 3.

Quercetin released from the PVA films in a similar profile to EGCG with a burst phase over the first day followed by a slow release over the next six days. There was no significant difference between samples. However, the amount of Quercetin released was much lower than for EGCG films, reaching only six to ten percent release by 1 day and only minor (but measurable) release after that time (Figure 4). One film loaded with Quercetin at 100 mg (no silver) was broken down after 7 days and showed 94% remaining drug in approximate agreement with released data showing less than 10% drug release.

### 3.3. Silver Release Study

Silver released from the PVA films more rapidly than EGCG and Quercetin released from the same films. For EGCG loaded films, silver nanoparticles released with a large burst phase for the 100 mg EGCG films but a smaller burst phase for the 75 mg EGCG loaded films (Figure 5). After five hours the release rate for both the 75 and 100 mg EGCG films was steady and sustained, reaching full release for the 100 mg films (measured at over 100%) at 7 days and 66% released for the 75 mg films. These release values were significantly different on day 3 and day 7 but not on day 2. Silver released from Quercetin loaded films without a burst phase but rather a steady release of approximately 55% by day 3 and a minor but measurable release by day 7 (Figure 6). There was no significant difference in release rates between the 100 mg and 200 mg (to 500 mg PVA) Quercetin-loaded films.

### 3.4. Fractional Inhibitory Concentration Index (FICI) Determinations

The minimal inhibitory concentrations (MIC) for EGCG, Tannic acid, and Quercetin against MRSA and *E. coli* were in the range of 50 to 100 μg/mL as seen in Table 3. The MIC for silver nanoparticles was much lower at 2.5 μg/mL. When placed in combination against these bacteria, the FICI values for all combinations were less than 0.5 (range 0.14 to 0.43) establishing that these combinations of antibiotics are synergistic. The same experiments were run using silver nitrate in place of silver nanoparticles. The MIC of silver nitrate against both bacteria was the same as for nanoparticles at 2.5 μg/mL. The FICI values against MRSA were 0.5 (EGCG), 0.75 (Tannic acid), and 0.62 (Quercetin). For *E. coli* the FICI values were 0.31 (EGCG), 0.56 (Tannic acid), and 0.56 (Quercetin). Therefore, silver nitrate is fully synergistic with EGCG for both bacteria and partially synergistic for Tannic acid and Quercetin (FICI of 0.75 or less).

### 3.5. In-PVA Gel Antibacterial Testing

When PVA films were placed in a small volume of broth containing bacteria, they swelled and formed robust gels, mimicking application to a wound with exudate. Films containing EGCG or Quercetin at two concentrations with or without silver nanoparticles along with control (no drug) or silver nanoparticles alone were compared for their ability to inhibit bacterial growth at 24 h using either MRSA or *E. coli*. Experiments were performed on three separate occasions, and all showed the same effect. The mean value of the three is shown, but because of different initial inoculum counts, error bars are not included.

### 3.6. EGCG with MRSA

After 24 h, the untreated bacteria proliferated from approximately 10^6^ CFU/mL to approximately 10^10^ CFU/mL. All films containing agents inhibited such growth (Figure 7A). The single agents alone (silver nanoparticles or EGCG) reduced bacterial growth to the order of 10^8^ to 5 × 10^6^ CFU/mL, but the combinations of EGCG and silver nanoparticles inhibited bacterial growth, restricting it below the original 10^6^ CFU/mL level with the higher EGCG loaded film killing all bacteria. The order of inhibition was EGCG100 + AGNP > EGCG75 + AGNP > EGCG100 > EGCG75 > AGNP.

### 3.7. EGCG with E. coli

*E. coli* proliferated from an initial inoculum of approximately 10^5^ CFU/mL to 5 × 10^9^ CFU/mL after 24 h (Figure 7B). Silver nanoparticles alone only produced mild inhibition of this growth but both concentrations of EGCG-loaded films strongly inhibited the proliferation of *E. coli* bacteria. The combination of EGCG and AGNP results in increased inhibition of *E. coli* growth as compared to either agent alone.

### 3.8. Quercetin with MRSA

MRSA grew rapidly over 24 h from approximately 10^5^ CFU/mL to approximately 10^8^ CFU/mL (Figure 7C). Films loaded with either Quercetin (100) or AGNP alone inhibited bacterial proliferation mildly, whereas the film loaded with the higher concentration of Quercetin inhibited proliferation strongly. Both films containing combinations of Quercetin and AGNP inhibited proliferation more than single agents alone. Indeed, the films loaded with the high (200) loading of Quercetin killed all bacteria.

### 3.9. E. coli with Quercetin

After 24 h the bacteria grew from an initial approximate inoculum of 10^5^/mL to approximately 10^8^ CFU/Ml (Figure 7D). The film loaded with the lower concentration of Quercetin and the film with silver nanoparticles alone inhibited bacterial growth mildly. Films loaded with combinations of silver nanoparticles and Quercetin inhibited bacterial proliferation more than films loaded with each agent alone.

## 4. Discussion

Poly vinyl alcohol hydrogels may be suitable wound dressings because they are comfortable, easy to remove, and may contain anti-infective drugs. However, PVA films manufactured using the 99% hydrolyzed PVA do not allow for any controlled degradation since they are largely insoluble. We have previously developed crosslinked degradable PVA films using the 88% hydrolyzed form by heat treatment in the presence of silver [7]. However, heat treatment complicates the manufacturing method, and may not be suitable for certain drugs and only occurs with silver ions present.

In this study, the gallate containing compounds, EGCG, Tannic acid, and Quercetin has been shown to allow for extensive crosslinking of PVA (88% hydrolyzed) as witnessed by concentration-dependent inhibition of dissolution with certain concentrations providing well-swollen (hydrogel) films (Figure 1 and Figure 2 and Table 2). The crosslinking is likely due to extensive hydrogen bonding between the gallate hydroxyl groups of tannic acid and the PVA hydroxyl groups on PVA and has been noted by many previous studies using 99% hydrolyzed PVA [16,17,18]. These previous studies sought to further strengthen PVA matrices rather than impacting dissolution in water. Similarly, tea polyphenols which contain many varieties of gallate-catechins including EGCG as well as caffeine [19,26,38] have been used to strengthen or impact the degradation of PVA films [19,20] via hydrogen bonding processes. There are limited studies using EGCG or Quercetin as singular agents directly in PVA films for mechanical or solubilization effects, although Quercetin has been used in PVA in combination with boron [39,40] and used as an antioxidant in PVA for food packaging [41]. The hydrogen bonding potential of Tannic acid and EGCG has been previously noted around drug solubilization of hydrophobic drugs [42,43,44].

The crosslinking effect of Tannic acid was short-lived and never lasted more than 24 h, but both EGCG and Quercetin provided swollen PVA gels at 4 days or more (Table 2). The reason for this less effective crosslinking by Tannic acid is unknown, but in terms of hydroxyl content in relation to molecular weight, then the order is EGCG > Quercetin > Tannic acid which may suggest an increased hydrogen bonding capacity of EGCG or Quercetin over Tannic acid. However, these different hydroxyl ratios are small, and the more likely influencing factor may the much larger molecular weight of Tannic acid (approximately 1700 dalton) as compared to EGCG (458) or Quercetin (302) so that steric effects may impact H bonding potential with the PVA polymer chains. Following the swelling studies, it was decided to use the EGCG (75) and EGCG (100) along with the Quercetin (100) and Quercetin (200) variants in further studies (all containing glycerol) since they allowed for swollen hydrogels with either partial or no degradation at four days (Table 2) for each crosslinking agent.

EGCG released well from the EGCG (75) and EGCG (100) PVA films with approximately 40% of the loaded drug releasing by day 3 which might provide high local concentrations of the agent on wounds for the critical first few days (Figure 3). It is likely that the unreleased fraction of the drug is more tightly bound in the PVA matrix. Quercetin released more slowly than EGCG, releasing between 6 and 10% of the loaded drug in the first three days. The minimal inhibitory concentrations of gallates against difficult-to-treat bacteria like *E. coli* and MRSA is of the order of 50–100 μg/mL (Table 3). All these films were cast in a 20 cm^2^ Petri dish so that if one-quarter of a film (5 cm^2^ with approximately 25 mg of gallate) was to be used for a small wound, then these rates of drug release would provide a huge excess of antibacterial agents based on these release rates. For example, the preferred care of diabetic wounds (which do have an average size of approximately 5 cm^2^) is debridement with saline washing and then methods to keep the wound moist [45]. So, if a 5 cm^2^ PVA-gallate film was placed on a wound with perhaps 1–2 mL of saline/exudate present in the swollen film, then the local gallate concentration would be easily sufficient as an anti-infective system.

Silver nanoparticles released more quickly from the PVA films than the gallates with approximately 50–60% of loaded silver released by two days for any films (Figure 5 and Figure 6). These levels of drug release from a 5 cm^2^ film (approximately 25–50 μg of total silver nanoparticles) placed on a wound with 2 mL of water/exudate might provide local concentrations of 6–12 μg/mL whereas the MIC of silver nanoparticles for *E. coli* and MRSA is 2.5 μg/mL (Table 3). Again, these release profiles theoretically provide sufficient drug alone for the inhibition of bacterial growth. Clearly the simultaneous release of EGCG (or Quercetin) with silver nanoparticles from these films should provide a huge excess of drug for extended periods on moist wounds.

Numerous studies have explored the antibiotic potential of gallate compositions as compared to compositions that also contain silver nanoparticles. These studies demonstrated an increased antibiotic potential over compositions containing silver or gallates alone [23,24,29,33,34]. However, these effects were not assessed by fractional inhibitory concentration index methods for synergy. In fact, although the word “synergy” is used frequently in this field, no studies measured FICI values and any reported increase in inhibitory effects were frequently small or did not include silver-alone measurements. In a study on wound closure, Kar et al. [44] demonstrated faster closure using EGCG with silver nanoparticle patches as compared to single agent patches but showed no increased inhibition in the growth of gram-positive or -negative bacteria using the combination systems. Similarly, Xiong et al. [34] described improved wound healing using EGCG plus silver nanoparticles over EGCG alone which was reported to arise from synergistic antibacterial activity (against *E. coli* and *Staph aureus*), yet improved antibacterial activity was less than 25% and silver alone was not measured. Badhwar et al. [33] demonstrated improved diabetic wound closure using Quercetin in combination with silver nanoparticles over each agent alone and mentioned synergistic effects against *E. coli* and *Staph aureus*, but the increased inhibition of bacterial growth was only measured in agar plates with only minor effects.

In this study, we have established strong antibacterial synergy among all three gallates (EGCG, Tannic acid, and Quercetin) and silver nanoparticles (Table 3). Similar results were found for the combined use of gallates with silver nitrate, whereby FICI values showed synergy (EGCG) or partial synergy (Quercetin and Tannic acid). The FICI values for silver nanoparticles were all well below 0.5 for both *E. coli* and MRSA.

These studies used the EGCG (75) and EGCG (100) or Quercetin (100) and Quercetin (200) loaded films as they had been shown to release agents well over 24 h and to allow for optimal crosslinked PVA hydrogel formation. These studies do not allow for determination of synergy but offer a suitable method for comparing the antibiotic potential of the films with individual agents or combinations of agents in a swollen aqueous setting like those found in a wound. These studies established that at all loadings of EGCG, Quercetin, and silver nanoparticles that the combined use of silver with EGCG or Quercetin provided increased inhibition of bacterial growth than films with single agents (Figure 7A–D). These data support the synergy findings established in the FICI studies (Table 2). All films had some inhibitory capacity over control films (PVA–glycerol, no drug) establishing the rationale for the use of PVA hydrogel films loaded with these drugs and especially with combinations of silver and gallates.

## Figures and Tables

**Figure 1 antibiotics-13-00312-f001:**
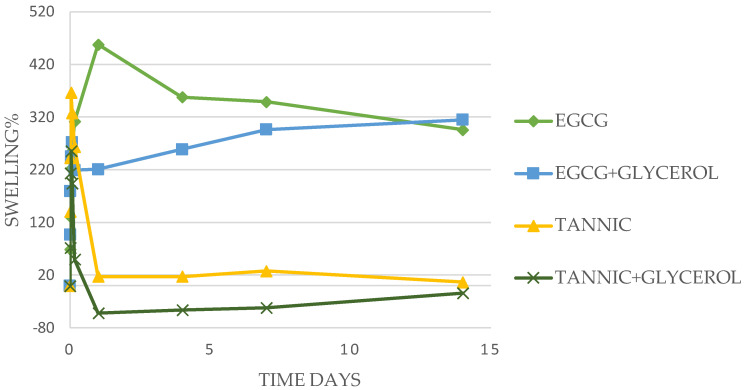
Swelling studies. PVA films (50 mg), with +/− 20% glycerol to PVA, and 0.04% silver nanoparticles, were tested for swelling in water. Initial films contained 100 mg EGCG in 500 mg PVA. Glycerol had no effect on swelling. Durable swelling was observed for EGCG, but only short-lived for tannic acid.

**Figure 2 antibiotics-13-00312-f002:**
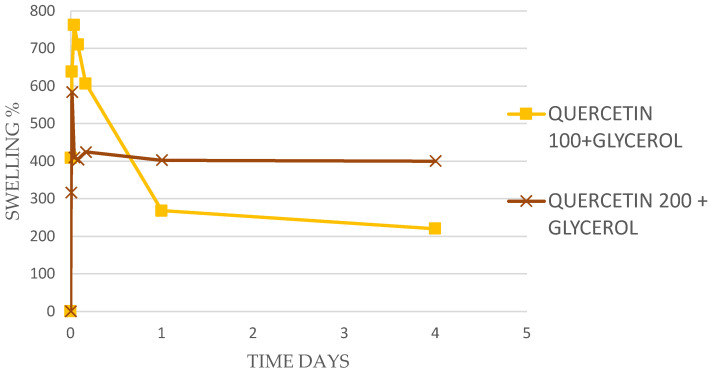
Swelling studies. PVA films (50 mg) each with 20% glycerol to PVA were tested for swelling in water. These films also contained silver nanoparticles at 0.02% (*w/w* to PVA). The initial films were loaded with either 100 mg or 200 mg of quercetin in 500 mg of PVA. It was noted that the films loaded with 100 mg and 200 mg of quercetin showed durable swelling.

**Figure 3 antibiotics-13-00312-f003:**
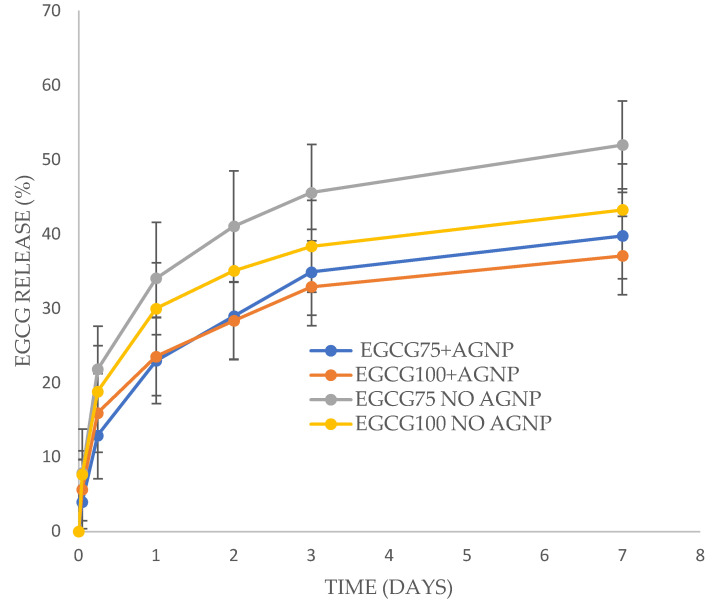
EGCG release from PVA films +/− silver nanoparticles (AGNP). Each film containing 75 or 100 mg of EGCG (to 500 mg of PVA in the initial film), along with 20% glycerol to PVA and 0.04% silver nanoparticles to PVA.

**Figure 4 antibiotics-13-00312-f004:**
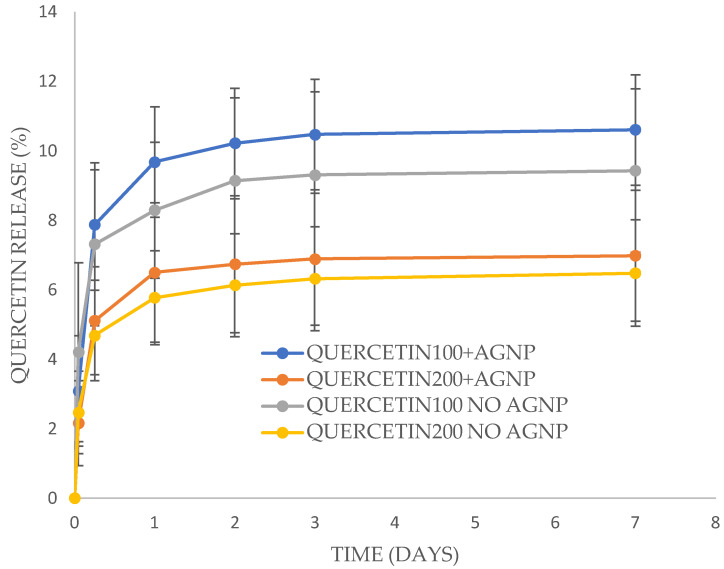
Quercetin release from PVA films +/− silver nanoparticles (AGNP). Each film containing 100 mg or 200 mg Quercetin (to 500 mg of PVA in the initial film), along with 20% glycerol to PVA and 0.02% silver nanoparticles to PVA.

**Figure 5 antibiotics-13-00312-f005:**
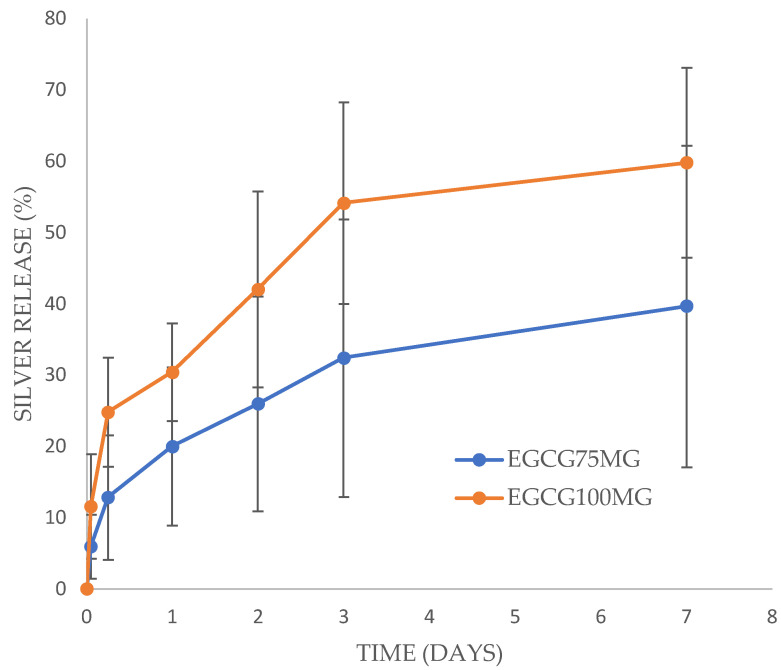
Silver nanoparticles released from EGCG-loaded PVA films. All films contained 20% glycerol and 0.04% silver nanoparticles. EGCG was loaded at either 75 mg (11% of PVA) or 100 mg (14% of PVA) to 500 mg PVA.

**Figure 6 antibiotics-13-00312-f006:**
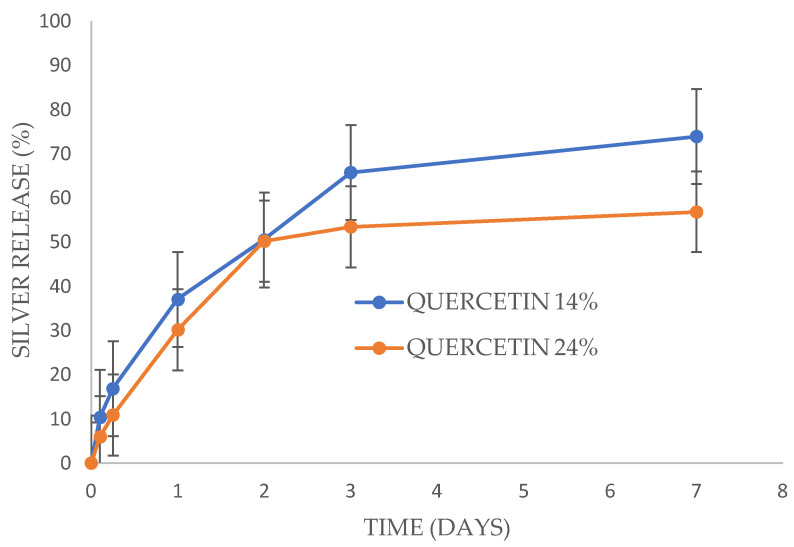
Silver nanoparticles released from Quercetin-loaded PVA films. All films contained 20% glycerol and 0.02% silver nanoparticles. Quercetin was loaded at either 100 mg (14% of PVA) or 200 mg (24% of PVA).

**Figure 7 antibiotics-13-00312-f007:**
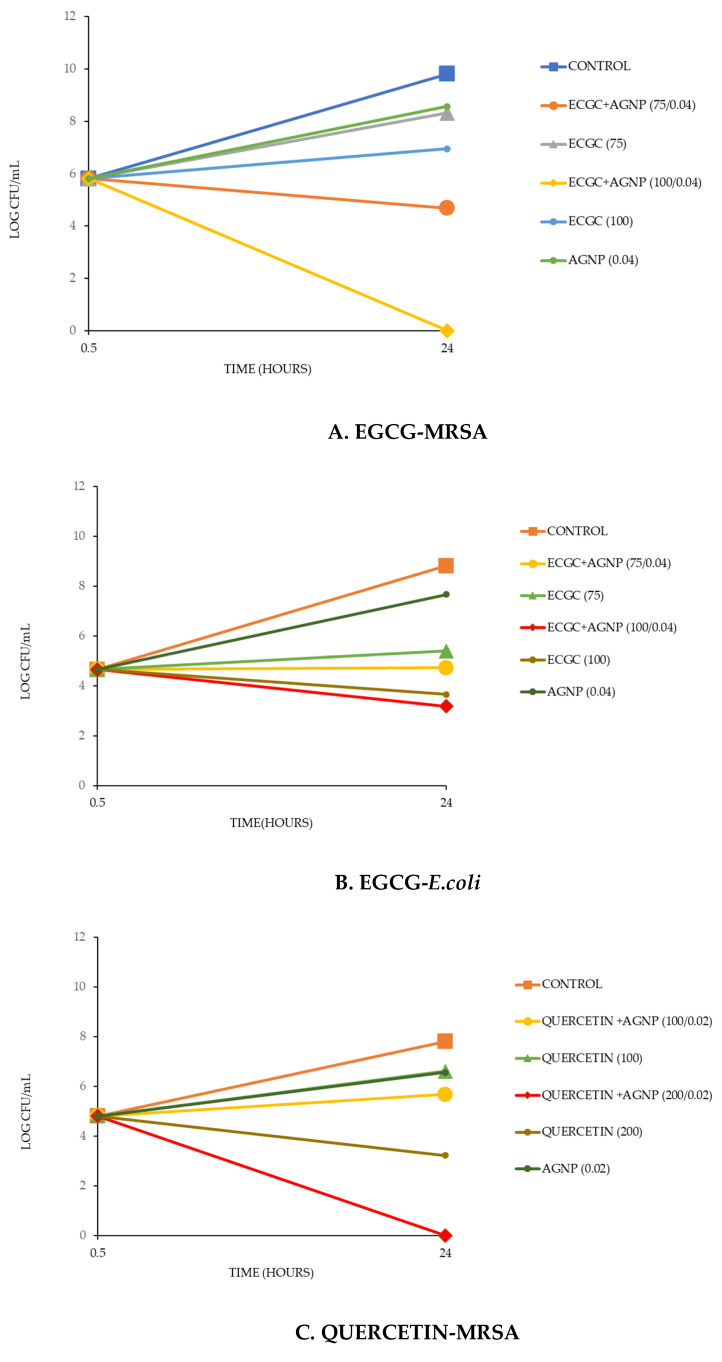
CFU bacterial studies. (**A**) 12.5 mg films soaked in 1 mL MRSA broth for 24 h, then CFU counted. All films contained 20% glycerol. Films contained EGCG at 75 mg or 100 mg (in 500 mg PVA) +/− silver nanoparticles (AGNP) at 0.04%. Control films contained PVA alone. (**B**) 12.5 mg films soaked in 1 mL *E. coli* broth for 24 h, then CFU counted. All films contained 20% glycerol. Films contained EGCG at 75 mg or 100 mg (in 500 mg PVA) +/− silver nanoparticles (AGNP) at 0.04%. Control films contained PVA alone. (**C**) 12.5 mg films soaked in 1 mL MRSA broth for 24 h, then CFU counted. All films contained 20% glycerol. Films contained Quercetin at 100 mg or 200 mg (in 500 mg PVA) +/− silver nanoparticles (AGNP) at 0.02%. Control films contained PVA alone. (**D**) 12.5 mg films soaked in 1 mL *E. coli* broth for 24 h, then CFU counted. All films contained 20% glycerol. Films contained Quercetin at 100 mg or 200 mg (in 500 mg PVA) +/− silver nanoparticles (AGNP) at 0.02%. Control films contained PVA alone.

**Table 1 antibiotics-13-00312-t001:** Composition of major films (weight %).

Title	PVA %	Gallate %	AGNP %	Glycerol %
EGCG 75	74	11	0.03	15
EGCG 100	71	14	0.03	14.3
Quercetin 100	71	14	0.014	14.3
Quercetin 200	63	25	0.012	12.5

**Table 2 antibiotics-13-00312-t002:** Swelling data were collected for PVA films all with 20% glycerol, containing different amounts of EGCG, tannic acid, or quercetin in an initial film with 500 mg of PVA. Films with EGCG or tannic acid also had 0.04% silver nanoparticles and quercetin films had 0.02% silver nanoparticles. The films were monitored over time, similar to Figure 1 and Figure 2. PVA-only films dissolved in under 2 h. Tannic acid films, despite initial swelling, did not stay swollen for 24 h. EGCG at 75 and 100 mg or quercetin at 100 or 200 mg (initially in 500 mg PVA films) which remained swollen for four days were used for further studies.

Swelling Studies (%)			
	Incubation Time		
500 mg PVA films with 20% glycerol (100 mg) and gallates	5 min	1 day	4 days
EGCG 50 mg	345	352	−54
EGCG 75 mg	286	370	91
EGCG 100 mg	271	220	260
EGCG 200 mg	112	152	216
EGCG 300 mg	84	92	88
EGCG 400 mg	114	120	126
EGCG 500 mg	90	92	94
Tannic 50 mg	151	180	35
Tannic 75 mg	147	82	57
Tannic 100 mg	147	74	27
Tannic 200 mg	135	45	35
Tannic 300 mg	154	59	55
Tannic 400 mg	159	69	79
Tannic 500 mg	179	70	76
Quercetin 50 mg	154	14	4
Quercetin 75 mg	798	256	170
Quercetin 100 mg	710	268	220
Quercetin 200 mg	404	402	400
Quercetin 300 mg	256	264	290

**Table 3 antibiotics-13-00312-t003:** Final FICI scores (MIC mean combination) for gallate synergy with silver nanoparticles in bacterial studies against gram-positive (MRSA) or gram-negative (*E. coli*) bacteria.

Gallate	MRSA			*E. coli*		
	MIC Alone(μg/mL)	MIC (Mean) Combination	St. Dev.	MIC Alone(μg/mL)	MIC (Mean) Combination	St. Dev.
EGCG	50	0.35	0.16	100	0.3	0.22
Tannic acid	50	0.197	0.057	50	0.43	0.18
Quercetin	50	0.147	0.05	100	0.43	0.11
Silver nanoparticles	2.5			2.5		

## Data Availability

Data are contained within the article.

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
