# Peer review of "Synergistic Antibacterial Effects of Gallate Containing Compounds with Silver Nanoparticles in Gallate Crossed Linked PVA Hydrogel Films"

_antibiotics, 2024, doi:10.3390/antibiotics13040312_

Round 1
Reviewer 1 Report
Comments and Suggestions for Authors
Jackson et al. reported the synergistic antibacterial effects of the combined use of natural product antibiotics gallates with silver nanoparticles. The manuscript shows the clear and professional use of the English language. The authors also created a comprehensive discussion section that delves into the antibacterial synergy between all three gallates and silver nanoparticles. Furthermore, the manuscript includes an up-to-date list of references. I only have a few suggestions regarding the clarity of some figures and wordings:
(1) Line 114: It is recommended to list the material of the 0.2 µm filtering disk in case its material makes a difference in the film swelling studies when readers try to repeat the results. Is it made of Nylon or other materials (PVC, MCE, PVDF, etc.)? What is the product number from Millipore?
(2) Table 2: "%" should be added to the numbers in this table indicating % swelling.
(3) Figures 7A, 7B, 7C, and 7D should all have figure titles to help readers understand the differences at first glance without needing to consult the figure captions.
Author Response
Reviewer 1.
- Line 114: It is recommended to list the material of the 0.2 µm filtering disk in case its material makes a difference in the film swelling studies when readers try to repeat the results. Is it made of Nylon or other materials (PVC, MCE, PVDF, etc.)? What is the product number from Millipore?
Response. We have now included this information in the new version of the manuscript.
- Table 2: "%" should be added to the numbers in this table indicating % swelling.
Response. We have added the % as requested.
- Figures 7A, 7B, 7C, and 7D should all have figure titles to help readers understand the differences at first glance without needing to consult the figure captions.
Response: We have given each Figure their own caption title.
Reviewer 2 Report
Comments and Suggestions for Authors
Figures font size, and font need to be consistent with the main text.
Characterization of the gel must be done: FTIR, XRD, FE-SEM,...
Drug release must be done in different temperature and drug concentration. --> calculate adsorption kinetics, isotherms, and thermodynamics.
Line 378-379: The claim" silver nps release more quickly..." do you have any proof?
How do you know your silver is nanoparticles? what is the conversion rate of silver salt (Ag+) to Ag0? --> UV-VIS?
Comments on the Quality of English LanguageMinor editing
Author Response
Reviewer 2
Figures font size, and font need to be consistent with the main text.
Response: We have corrected all fonts to Palatino size 10 throughout the text. We have changed all the figure font sizes to capitals and Palantino 10 to match the text. We agree they look better. One exception is Figure 7 which has lots of information in the legend and the figure wont take this font in capitals and size 10. We feel Figure 7 presents very well as is but we will discuss this with the editor
Characterization of the gel must be done: FTIR, XRD, FE-SEM,...
Response: Our group is usually very keen on pharmaceutical characterization of polymeric formulations. However, the problem here is that we create a dry film but the functional form is a gel. We cannot run such characterizations on the wet swollen form used in these studies. We did contemplate characterizing the dry form but we felt any data would not offer the reader any further information or improve understanding. Instead it might lead to confusion as to why the data was included. We feel the novel findings in this study are simply presented and the story is easy to grasp. We hope the reviewer understands.
Drug release must be done in different temperature and drug concentration. --> calculate adsorption kinetics, isotherms, and thermodynamics.
Response: The main factor for formulation design in this study was to provide a slow degrading hydrogel. Tannic acid films quickly degraded and were not used further. We only used EGCG (75,100) and Quercetin (100, 200) films that allowed for swollen PVA hydrogels for extended periods without degradation. We only used very low concentrations of silver nanoparticles as the FICI data showed effective antibacterial effects at these concentrations. Silver nanoparticles are very expensive so keeping the concentration as low as possible is important. We therefore concentrated our drug release experiments on these formulations as being most relevant to a final clinically relevant formulation. We ran the release studies at 37oC only as this is relevant to in vivo applications. We have now included an explanation for our release study design in the Results section.
Line 378-379: The claim" silver nps release more quickly..." do you have any proof?
Response: We have included the appropriate figure numbers in the text after that statement. (Figure 5 and 6)
How do you know your silver is nanoparticles? what is the conversion rate of silver salt (Ag+) to Ag0? --> UV-VIS?
Response: in much of our previous work we used silver nitrate and found that under reduced conditions in heated PVA films, that there was significant conversion to silver nanoparticles. Many research papers (all referenced in this manuscript) showed that EGCG, Quercetin or Tannic acid promoted such conversion. In this study we did not use silver salts but intact silver nanoparticles to avoid unquantifiable conversion effects or gallate surface capping effects.
Reviewer 3 Report
Comments and Suggestions for Authors
This manuscript aimed to develop silver-based antiseptic wound dressings by incorporating silver nanoparticles into gallate (EGCG)-containing PVA (88% hydrolyzed) cast films. Gallates functioned as both PVA cross-linking agents and potential synergistic antibiotics when combined with silver nanoparticles. The evaluation encompassed crosslinking efficiency, silver and gallates release, synergy against bacteria, and antibacterial effects in hydrogel films. The findings suggest that EGCG can yield stable PVA hydrogel films with potent synergistic antibiotic effects when combined with silver nanoparticles. However, there are a few aspects that need to be promoted or clarified. Some comments on this work are shown below:
1. Standardization of figure captions is necessary. Captions in Figures 1 and 2 are lowercase, while those in other figures are uppercase.
2. Regarding swelling studies in Figures 1 and 2, the initial testing time intervals were short, and the entire testing period was prolonged. This may result in dense data points before 1 day, hindering curve trend observation. Redrawing the figures is recommended.
3. Error bars are absent in tables and figures. It's crucial to specify the number of experiment repetitions and add error bars for clarity.
4. In Drug Release Studies (Figures 3-6), it's unclear whether the release curve was obtained by changing fresh buffer daily or soaking in the same buffer for 7 days. Detailed experimental steps and clarification on regular removal intervals are necessary. Additionally, it's important to specify if silver ions and EGCG were tested in the same collected samples.
5. MRSA and E. coli were selected in this study to assess the antibacterial efficacy of the film. Why Gram-positive bacteria are drug-resistant bacteria and Gram-negative bacteria are standard strains?
6. Data processing methods need enhancement. For CFU antibacterial studies, displaying results with a histogram could offer better clarity. Additionally, clarification is needed regarding the composition of the control films. Although the figure legend states that the control films contained AgNPat 0.04% alone, the appearance of the control group in the figure suggests that it resembled a film containing only PVA. This discrepancy should be addressed to ensure accurate interpretation of the experimental results.
7. And in the Figure legend, author claimed that “Control films contained AgNP at 0.04% alone”. According to the Figure, The control group looked like a film containing only PVA.
8. Biocompatibility of the antibacterial film and potential biological toxicity require evaluation.
9. Why choose silver nanoparticles instead of other antimicrobial nanoparticles, such as gold nanoparticles? Literature indicates superior biocompatibility of antibacterial gold nanoparticles.
Comments on the Quality of English LanguageOverall, the manuscript is well written. The language used in the article is generally concise and easy to understand.
Author Response
Reviewer 3.
This manuscript aimed to develop silver-based antiseptic wound dressings by incorporating silver nanoparticles into gallate (EGCG)-containing PVA (88% hydrolyzed) cast films. Gallates functioned as both PVA cross-linking agents and potential synergistic antibiotics when combined with silver nanoparticles. The evaluation encompassed crosslinking efficiency, silver and gallates release, synergy against bacteria, and antibacterial effects in hydrogel films. The findings suggest that EGCG can yield stable PVA hydrogel films with potent synergistic antibiotic effects when combined with silver nanoparticles. However, there are a few aspects that need to be promoted or clarified. Some comments on this work are shown below:
- Standardization of figure captions is necessary. Captions in Figures 1 and 2 are lowercase, while those in other figures are uppercase.
Response: We have standardized the figures 1 to 7 so they are all uppercase and the same font and size.
- Regarding swelling studies in Figures 1 and 2, the initial testing time intervals were short, and the entire testing period was prolonged. This may result in dense data points before 1 day, hindering curve trend observation. Redrawing the figures is recommended.
Response: There were lots of time point tests in the 24 hours because PVA alone fully dissolves in less than 2 hours. We did not know if the gallate treated PVA would degrade in the first 24 hours so we took lots of time interval points in the first day. However, we soon realized that some films showed extensive crosslinking and slow degradation effects. In fact these longer time point data are the important points. For example we eliminated tannic acid from further studies because all films degraded by 24 hours. The data shown in Figures 1 and 2 demonstrate the extended stability of the optimal EGCG or Quercetin crosslinked PVA films and we feel the early time points are not as important as the longer ones.
- Error bars are absent in tables and figures. It's crucial to specify the number of experiment repetitions and add error bars for clarity.
Response: All release curves were performed using triplicate samples. We have now included error bars on the mean values in Figures 3, 4, 5 and 6. The swelling studies were repeated on two separate occasions and gave similar results. These swelling studies required approximately 20 different samples, each with 8-10 time points so it was impossible to handle triplicate samples for each condition. The objective of these swelling studies was look for concentration (gallate) dependent crosslinking as witnessed by extended swelling over time and this was easily observed for EGCG and Quercetin without the need for triplicate experiments. We have now explained this aspect in the Results section with new text. The CFU experiments in Figures 7 A-D. were repeated on three separate occasions and gave the same overall results in terms of order of bacterial growth per condition. However, because the original CFU starting counts were quite different from experiment to experiment (note this is a log scale with big numbers), the actual values of the CFU’s for each condition were different making error bars inappropriate. We have explained this in the Results section of the new version of the manuscript.
- In Drug Release Studies (Figures 3-6), it's unclear whether the release curve was obtained by changing fresh buffer daily or soaking in the same buffer for 7 days. Detailed experimental steps and clarification on regular removal intervals are necessary. Additionally, it's important to specify if silver ions and EGCG were tested in the same collected samples.
Response. Our apologies for the lack of clarity in the Methods section. The buffer was fully changed at each sample point and the same sample was used for separate HPLC analysis of gallates and ICP measurement of silver. We have explained this in the new version of the Methods.
- MRSA and E. coli were selected in this study to assess the antibacterial efficacy of the film. Why Gram-positive bacteria are drug-resistant bacteria and Gram-negative bacteria are standard strains?
Response: Non resistant SA is quite easy to treat. Of more interest to the microbiology community are novel ways of counteracting difficult- to- treat MRSA as the gram positive species. E. coli is a notoriously hard- to- treat gram negative bacteria. These bacteria (MRSA and E.coli) are the standard pairing used in most antibacterial studies because they represent common difficult- to- treat species of either gram discipline. We have included an explanation of bacterial choices in the new version of the manuscript.
- Data processing methods need enhancement. For CFU antibacterial studies, displaying results with a histogram could offer better clarity. Additionally, clarification is needed regarding the composition of the control films. Although the figure legend states that the control films contained AgNPat 0.04% alone, the appearance of the control group in the figure suggests that it resembled a film containing only PVA. This discrepancy should be addressed to ensure accurate interpretation of the experimental results.
Response: We thank the reviewer for noting the alternative use of histograms. We did try plotting this way but because there are so many samples we felt the legend description of each sample was difficult to connect to each histogram column and the figures looked very “messy”. We feel the essential order of the inhibition of bacterial growth is adequately described as graphs and are easy to read. We will change this to histograms if the reviewer further requests this. Our apologies for the mistake in Figure 7. Control films were PVA alone. We have corrected this in the new figure titles.
- And in the Figure legend, author claimed that “Control films contained AgNP at 0.04% alone”. According to the Figure, The control group looked like a film containing only PVA.
Response: The reviewer is correct. We have changed the title to each figure to explain that the control films were PVA alone.
- Biocompatibility of the antibacterial film and potential biological toxicity require evaluation.
Response: We believe that biocompatibility is the next step for this research. In fact, planned wound healing experiments will reveal much of this information. It is well known that PVA is a fully biocompatible polymer and has been used extensively in hydrogel research. Silver is commonly used in wound dressings and the gallates are orally consumed in high concentrations (e.g. green tea) and used in multiple dermal compositions so we feel it is likely that the compositions described in this paper will be fully biocompatible.
- Why choose silver nanoparticles instead of other antimicrobial nanoparticles, such as gold nanoparticles? Literature indicates superior biocompatibility of antibacterial gold nanoparticles.
Response: Silver nanoparticles are the standard of care in anti-infective wound healing products (e.g. Acticoat tm). We hoped to improve this existing antibacterial effect of silver nanoparticles by showing synergy with gallates so that even lower concentrations of silver might be required for the same antibacterial potential. It would be interesting to replicate these studies using gold nanoparticles but it is likely the experiments would be very expensive.
Round 2
Reviewer 2 Report
Comments and Suggestions for Authors
N/A